# Factors Associated with COVID-19 Vaccine Booster Hesitancy: A Retrospective Cohort Study, Fukushima Vaccination Community Survey

**DOI:** 10.3390/vaccines10040515

**Published:** 2022-03-26

**Authors:** Makoto Yoshida, Yurie Kobashi, Takeshi Kawamura, Yuzo Shimazu, Yoshitaka Nishikawa, Fumiya Omata, Tianchen Zhao, Chika Yamamoto, Yudai Kaneko, Aya Nakayama, Morihito Takita, Naomi Ito, Moe Kawashima, Sota Sugiura, Kenji Shibuya, Shingo Iwami, Kwangsu Kim, Shoya Iwanami, Tatsuhiko Kodama, Masaharu Tsubokura

**Affiliations:** 1Faculty of Medicine, Teikyo University School of Medicine, Itabashi Campus, Tokyo 173-8605, Japan; 19m101097pr@stu.teikyo-u.ac.jp; 2Department of Radiation Health Management, Fukushima Medical University School of Medicine, Fukushima 960-1247, Japan; yurie-s@umin.ac.jp (Y.K.); yuzo0806@fmu.ac.jp (Y.S.); cho1230@fmu.ac.jp (T.Z.); takita-ygc@umin.net (M.T.); itonaomi@fmu.ac.jp (N.I.); m181038@fmu.ac.jp (M.K.); 3Department of General Internal Medicine, Hirata Central Hospital, Fukushima 963-8202, Japan; ynishikawa-tky@umin.ac.jp (Y.N.); omata-thk@umin.ac.jp (F.O.); 4Proteomics Laboratory, Isotope Science Center, The University of Tokyo, Tokyo 113-0032, Japan; kawamura@lsbm.org (T.K.); nakayama-a@lsbm.org (A.N.); 5Laboratory for Systems Biology and Medicine, Research Center for Advanced Science and Technology, The University of Tokyo, Tokyo 153-8904, Japan; kaneko.yudai@mbl.co.jp (Y.K.); kodama@lsbm.org (T.K.); 6Japan Red Cross Society Fukushima Hospital, Fukushima 960-8530, Japan; chika.y.9112@gmail.com; 7Medical & Biological Laboratories Co., Ltd., Tokyo 105-0012, Japan; 8Medical Governance Research Institute, Tokyo 108-0074, Japan; bluebig0287@g.ecc.u-tokyo.ac.jp; 9Soma Medical Center of Vaccination for COVID-19, Fukushima 976-8601, Japan; shibuya@tkfd.or.jp; 10Tokyo Foundation for Policy Research, Tokyo 106-6234, Japan; 11Interdisciplinary Biology Laboratory (iBLab), Division of Biological Science, Graduate School of Science, Nagoya University, Nagoya 464-8602, Japan; iwami.iblab@bio.nagoya-u.ac.jp (S.I.); kim.iblab@bio.nagoya-u.ac.jp (K.K.); iwanami.iblab@bio.nagoya-u.ac.jp (S.I.); 12Minamisoma Municipal General Hospital, Minamisoma 975-0033, Japan

**Keywords:** antibody, BNT162b2, coronavirus disease 2019, severe acute respiratory syndrome coronavirus 2, vaccine hesitancy, vaccine booster, pandemic

## Abstract

This was a retrospective cohort study, which aimed to investigate the factors associated with hesitancy to receive a third dose of a coronavirus disease 2019 (COVID-19) vaccine. A paper-based questionnaire survey was administered to all participants. This study included participants who provided answers in the questionnaire about whether they had an intent to receive a third dose of a vaccine. Data on sex, age, area of residence, adverse reactions after the second vaccination, whether the third vaccination was desired, and reasons to accept or hesitate over the booster vaccination were retrieved. Among the 2439 participants, with a mean (±SD) age of 52.6 ± 18.9 years, and a median IgG-S antibody titer of 324.9 (AU/mL), 97.9% of participants indicated their intent to accept a third vaccination dose. The logistic regression revealed that participants of a younger age (OR = 0.98; 95% CI: 0.96–1.00) and with a higher antibody level (OR = 2.52; 95% CI: 1.27–4.99) were positively associated with hesitancy over the third vaccine. The efficacy of the COVID-19 vaccine and concerns about adverse reactions had a significant impact on behavior regarding the third vaccination. A rapid increase in the booster dose rate is needed to control the pandemic, and specific approaches should be taken with these groups that are likely to hesitate over the third vaccine, subsequently increasing booster contact rate.

## 1. Introduction

Coronavirus disease 2019 (COVID-19) was first confirmed in China in December 2019, causing pneumonia and other respiratory diseases, and rapidly spreading worldwide, with a cumulative total of 400,846,690 confirmed cases and 5,764,834 deaths as of 9 February 2022 [1]. Given its high prevalence, an effective vaccine would play a critical role in preventing the spread of COVID-19 [2]. Many efforts have been made to develop vaccines, including the BNT162b2 (Pfizer/BioNTech, New York, NY, USA), mRNA-1273 (Moderna, Cambridge, UK), Ad26.COV2.S (Janssen/Johnson & Johnson, New Brunswick, NJ, USA), and BBIBP-CorV (Sinopharm, Beijing, China), which were approved by the World Health Organization [3]. However, vaccine hesitancy has been observed worldwide and is a major obstacle in controlling the COVID-19 pandemic [4,5,6,7]. In addition, there are concerns that the neutralizing activity of a vaccine may decrease over time, even in individuals who have received two doses of the vaccine. It was reported that six months after the administration of the second dose of the BNT162b2 vaccine, the humoral response was substantially reduced [8]. In several jurisdictions, booster vaccinations have been authorized by the regulatory authorities [9]. The booster vaccination dose level is the same in the BNT162b2 vaccine and half the level in the mRNA-1273 vaccine compared to the dose in the primary vaccination series. As described above, reducing the number of unvaccinated individuals, as well as minimizing vaccine hesitancy to facilitate more people receiving a third dose, is an extremely important measure in controlling the COVID-19 pandemic.

Previous studies examined the factors associated with vaccine hesitancy. Ethnicity, work status, religion, politics, sex, age, education, and income were reported to be factors associated with COVID-19 vaccine hesitancy [10,11]. However, information on the factors affecting COVID-19 booster vaccine hesitancy among fully-vaccinated individuals is limited [12,13,14,15,16].

Fukushima Prefecture experienced a surge in COVID-19 cases between August 2021 and January 2022, wherein a total of 17,327 cases, equivalent to 194 cases per 100,000 people, were reported by 9 February 2022 [17]. BNT162b2 vaccinations were offered beginning March 2021 to healthcare workers, older adults, people with comorbidities, and the general population. A total of 81.7% of the population aged ≥12 years who were eligible for vaccination received second doses by 2 February 2022. A third vaccination dose was offered from1 December 2021 to healthcare workers and residents of geriatric facilities who had been vaccinated with either the BNT162b2 vaccine or the mRNA-1273 vaccine at least six months earlier, and adults aged 65 years or older who had been vaccinated for at least eight months. In Japan, 3.96% of the population received a third dose by 3 February 2022, which is the lowest among the OECD member countries. In Fukushima Prefecture, 5.1% of the total population received a third dose as of 3 February 2022 [18]. In addition, certain disaster-affected populations in Fukushima have been continuously subjected to COVID-19 antibody titer monitoring to develop infection prevention measures accordingly [19,20]. This area had an advantage in that the residents were aware of the level of their own humoral immunity after vaccination and how it had evolved, allowing assessment of how this affected their behavior regarding the third vaccination.

The purpose of this study was to investigate factors associated with hesitancy over a third dose of the COVID-19 vaccine in order to increase the booster contact rate required to control the COVID-19 pandemic. This study was supported by the Japan Agency for Medical Research and Development and was conducted as part of the FVCS, a study carried out to evaluate antibody titer dynamics after the second dose of the BNT162b2 vaccine and the mRNA-1273 vaccine in rural communities in Japan.

## 2. Materials and Methods

### 2.1. Study Design and Population

A total of three participants received two doses of mRNA-1273 and all other participants received two doses of BNT162b2. All the participants who received mRNA-1273 were non-healthcare workers. The vaccination dose was the same for all the participants. The blood sampling was performed between 8 September and 8 October 2021. The mean (SD) of the interval days between the second dose of the vaccination and the serological tests was 180.5 (±32.2) days, ranging from 64 to 252 days. Participants were a predominantly homogenic Japanese group, recruited mainly from rural Fukushima Prefecture, including Ishikawa gun, Soma City, and Minamisoma City.

In December 2021, a paper-based questionnaire survey inquiring information regarding sex, age, area of residence, adverse reactions after the second vaccination, whether the third vaccination was desired, and reasons for wanting or not wanting the vaccination was distributed to all participants. Subsequently, the eligibility criterion for the study participants was that they provided an answer in the questionnaire to whether they had an intent to receive the third dose of the vaccine. Children aged ≥12 years were allowed to receive the COVID-19 vaccination; thus, we included 58 participants aged 12–19 years.

### 2.2. Questionnaire Survey about Booster Vaccine Acceptance

We conducted a questionnaire survey on the acceptance of the third vaccination. The questionnaire items concerning the reasons for acceptance or hesitancy towards the third vaccination could be answered in multiple ways. Papers on vaccine hesitancy [10,11] were used as reference to prepare the questionnaire survey. We also considered the opinion of the medical staff, local government staff, and researchers involved in the vaccination process.

### 2.3. Serological Assay

Blood sampling was conducted between 8 September and 8 October 2021, at each facility in the rural Fukushima Prefecture. Centrifugal separation of blood samples was performed at each facility, and serum samples were sent to Tokyo University. The levels of immunoglobulin G (IgG) antibodies against the severe acute respiratory syndrome coronavirus 2 (SARS-CoV-2) spike (S1) protein were measured at Tokyo University. Serological assays were performed using the CLIA assay with iFlash 3000 (YHLO Biotech, Shenzhen, China) and iFlash-2019-nCoV series (YHLO Biotech) as reagents. The cut-off value for the IgG antibody against the S1 protein was ten arbitrary units per milliliter (AU/mL), which was the standard cut-off value. The testing process complied with the official guidelines [21,22]. The results of the serological assay were shared with the participants in November 2021.

### 2.4. Statistical Analysis

We compared the characteristics of participants according to their willingness to receive a third COVID-19 vaccine dose. We conducted a chi-square test for sex, municipality, and adverse reactions; a *t*-test for age; and a Wilcoxon rank-sum test for the IgG antibody titer. The reasons to accept or hesitate over the third vaccination dose were tabulated separately. A logistic regression was used to assess the relationship between third vaccine hesitancy and age, sex, number of adverse reactions after the second vaccination, whether an IgG-S antibody titer was higher or lower than the median (324.85 AU/mL), and place of residence. Since the antibody titers were affected by days since the vaccination (Correlation coefficient was −0.38), we used the antibody titer only as the model from the perspective of multicollinearity. A logistic regression was used to assess the relationship between third vaccine hesitancy and age, sex, fever above 37.5 °C, fatigue, headache, joint pain, nausea, whether an IgG-S antibody titer was above or below the median (324.85 AU/mL) after the second vaccination, and place of residence. We considered *p* values of 0.05 or less to be significant. The statistical software STATA IC (version 15; Lightstone, DL, College Station, TX, USA) was used for all analyses.

## 3. Results

Among the 2439 participants, with a mean (±SD) age of 52.6 ± 18.9 years, median IgG-S antibody titer of 324.9 (AU/mL), and of whom 41.7% were male, 97.9% indicated their intent to accept the third vaccination dose. The mean age was significantly lower (*p* = 0.001), while the median IgG-S score was significantly higher in the vaccine-hesitant group (*p* < 0.001). The frequency of nausea as an adverse reaction after the second vaccination was also significantly higher in the vaccine-hesitant group (*p* < 0.001) (Table 1).

The main reasons for accepting the third vaccination were as follows: “necessary for infection control” (81.8%); “vaccines are highly effective” (47.3%); and “adverse reactions are not a major concern” (27.1%). The main reasons to hesitate over the third vaccination were as follows: “worried about adverse reactions” (57.7%); “two doses of vaccine are sufficient” (25.0%); and “efficacy is unknown” (19.2%) (Table 2). Concerns about adverse reactions and efficacy of the vaccine were the main reasons for hesitance towards a third vaccination. In contrast, “necessary for infection control” was the main reason for acceptance of the third vaccination, in addition to a lack of concern about the efficacy and adverse reactions.

The results of the logistic regression reveal that the categories of younger age (OR = 0.98; 95% CI: 0.96–1.00) and higher antibody level (OR = 2.52; 95% CI: 1.27–4.99) are predominantly positively associated with third vaccine hesitancy. In contrast, sex and the number of adverse reactions at the second vaccination are not significantly associated with third vaccine hesitancy (Table 3). Furthermore, the results of the multilevel model analysis show that younger age and the experience of adverse reactions such as nausea are significantly associated with third vaccine hesitancy (*p* < 0.001) (Appendix A).

## 4. Discussion

Reducing vaccine hesitancy is important in controlling the COVID-19 pandemic. In this study, we conducted a retrospective cohort study by distributing questionnaires in December 2021 to populations with measured IgG-S antibody titers for COVID-19 between 8 September and 8 October 2021 in rural areas of Japan to investigate the factors associated with third vaccine hesitancy.

In this study, we found that participants who acquired relatively high humoral immunity after two doses of vaccination were associated with third vaccine hesitancy. Categories such as younger age (OR = 0.98; 95% CI: 0.96–1.00) and previous antibody titers higher than the median (OR = 2.52; 95% CI: 1.27–4.99) were positively associated with vaccine hesitancy. This result is consistent with previous reports that younger individuals are less likely to accept vaccines in the primary vaccination series as well as the booster dose [5,12,23,24,25,26]. Since antibody levels decrease over time [8], it is not guaranteed that the third vaccination is unnecessary when the antibody level is higher than the median. Thus, it is essential to provide information about the importance of a third dose and discuss how to interpret the results of antibody levels.

Approximately 97.9% of FVCS participants who had their own antibody titers measured accepted a third vaccination. This result was higher than other reports of third vaccination acceptance [13,14,15]. It is also suggested that education and policy-based interventions for vaccinations should be implemented [10,27,28,29,30]. Continuous antibody testing, careful explanation of test results on paper, and information sharing through meetings among medical staff may have contributed to the high vaccination acceptance rate.

The efficacy of the COVID-19 vaccine and concerns about adverse reactions had a significant impact on the reasons for behavior regarding the third vaccination. Previous studies have shown that vaccine efficacy, safety, and a lack of concern regarding adverse reactions increase vaccine acceptance [10,27,31,32,33,34,35,36,37,38]. It is important to continue to consider these factors when designing communication about vaccines.

A limitation of this study was that the participants were recruited in a disaster-affected area, using a unique network developed with more than 10 years of disaster recovery through a multi-sectoral partnership. These participants were interested in vaccines and health issues, and voluntarily signed up to participate in the FVCS. This may have introduced a sampling bias and may not be conclusive of other populations. In addition, a few people hesitated over the third vaccination, and only 58 of the approximately 2500 participants were under the age of 18, which made the statistical analysis challenging. In addition, it was not possible to eliminate potential confounding factors such as media preferences, education, and income. Moreover, we were unable to collect information on the occupations that might affect vaccine acceptance. Furthermore, nearly everyone had received the BNT162b2 vaccine, making it difficult to verify whether vaccine acceptance differed between the different types of vaccines. Finally, considering the development of vaccination strategies, vaccine equity and immune competition are also key determinants [39].

## 5. Conclusions

In this study, we found that younger age groups and participants with antibody levels higher than the median were associated with third vaccine hesitancy. A rapid increase in the booster dose rate is needed to control the virus, and specific approaches should be taken in these groups that are likely to hesitate over the third vaccine. Further research on factors associated with third vaccine hesitancy should also be explored.

## Figures and Tables

**Table 1 vaccines-10-00515-t001:** Characteristics of the participants based on willingness to accept the third COVID-19 vaccine dose *n* (%) (*n* = 2439).

	Accept Vaccine	Hesitate over Vaccine	Total
Age ** (mean [SD])	52.8 [18.9]	44.0 [19.0]	52.6 [19.3]
Sex			
Male	998 (41.8)	20 (38.5)	1018 (41.7)
IgG antibody titer *** (median (25th 75th))	319.6 (171.7–571.3)	549.9 (446.0–724.1)	324.9 (173.2–576.3)
Municipality *			
Hirata	1343 (56.3)	37 (71.2)	1380 (56.6)
Soma	464 (19.4)	5 (9.6)	469 (19.2)
Minamisoma	580 (24.3)	10 (19.2)	590 (24.2)
Adverse reaction			
Local pain	1358 (57.0)	24 (46.2)	1382 (56.8)
Over 37.5 °C fever *	676 (28.4)	21 (40.4)	697 (28.7)
Fatigue	1195 (50.2)	28 (53.9)	1223 (50.3)
Headache **	646 (27.1)	22 (42.3)	668 (27.4)
Joint pain	728 (30.6)	18 (34.6)	746 (30.7)
Diarrhea	53 (2.2)	2 (3.9)	55 (2.3)
Nausea ***	87 (3.7)	87 (17.3)	96 (3.9)
Dizziness	102 (4.3)	3 (5.8)	105 (4.3)

We conducted a chi-square test for sex, municipality, and adverse reactions; a *t*-test for age; and a Wilcoxon rank-sum test for IgG antibody titer. ***: *p*-value < 0.001. **: *p*-value < 0.05. *: *p*-value < 0.1.

**Table 2 vaccines-10-00515-t002:** Reasons to accept or hesitate over a third COVID-19 vaccine dose (*n* = 2439).

	*n* (%)
Reasons to accept the third vaccination (*n* = 2387)	
Necessary for infection control	1952 (81.8)
Vaccines are highly effective	1128 (47.3)
Adverse reactions are not a major concern	647 (27.1)
The second vaccination is not effective enough	494 (20.7)
Considering the number of people infected with the COVID-19	325 (13.6)
Vaccine is safe	311 (13)
Access to vaccination sites is good	300 (12.6)
Low antibody levels	280 (11.7)
I have a chronic disease	181 (7.6)
No work shift	118 (4.9)
Reasons to hesitate over the third vaccination (*n* = 52)	
Worried about adverse reactions	30 (57.7)
Two doses of vaccine are sufficient	13 (25.0)
Efficacy is unknown	10 (19.2)
Worried about long term effects	8 (15.4)
Too much trouble	4 (7.7)
The number of people infected with COVID-19 is small.	4 (7.7)
Antibody titers are high	3 (5.8)
Other methods of infection control are sufficient	2 (3.8)
I can’t take a day off work or school.	1 (1.9)
I don’t have a chronic disease.	0 (0.0)

**Table 3 vaccines-10-00515-t003:** Logistic regression analysis to identify the variables influencing vaccine hesitancy.

	B (se)	OR (95% CI)	*p*-Value
Age	−0.021	0.98 (0.96–1.00)	0.030
Sex (base: male)	0.066	1.07 (0.60–1.92)	0.82
Number of whole-body adverse reactions	0.049	1.05 (0.87–1.27)	0.61
IgG antibody titer	0.920	2.52 (1.27–4.99)	0.008
Municipality (base: Hirata)			
Soma	−1.060	0.35 (0.13–0.91)	0.031
Minamisoma	−0.580	0.56 (0.28–1.15)	0.113

Whole-body adverse reactions include fever over 37.5 °C, fatigue, headache, joint pain, diarrhea, nausea, and dizziness.

## Data Availability

Not applicable.

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
