# Peer review of "Factors Associated with COVID-19 Vaccine Booster Hesitancy: A Retrospective Cohort Study, Fukushima Vaccination Community Survey"

_vaccines, 2022, doi:10.3390/vaccines10040515_

Round 1

Reviewer 1 Report

This paper presents the statistical results of an investigation on the willingness   to accept COVID19 Booster vaccination. The results  of the paper have been   obtained by the responses to a questionnaire to inquire also. The  study is well organized and it is based on real data.  The results on the response can be useful to risk managers in charge of organizing vaccination programs. Therefore, I recommend publication after some revisions aimed at referring their investigation also to a general framework of the complex dynamics of the virus in human societies. In details, the authors should be invited to care about the following topics:

  1. A paper on a closed topic has been recently published on Vaccines. The authors should consider this paper and discuss similarities and differences.

Folcarelli, L., Del Giudice, G.M., Corea, F., Angelillo, I.F., Intention to Receive the COVID-19 Vaccine Booster Dose in a University Community in Italy, Vaccines 10(2), 146.

  1. The use of the statistics to different aspects related to the interpretation of the contents of the paper is mainly left to the authors’ claim, but the authors should make an effort to address the use which goes beyond crisis management, but also to various aspects related to the pandemics viewed as a complex systems in a globally connected world. In particular, there are studies generated by the following article:

   https://www.worldscientific.com/doi/pdf/10.1142/S0218202520500323

which investigates the role of the immune competition on the shape of subsequent waves after the first wave.

This role is activated by vaccination programs etc.

Author Response

Response to reviewer 1

This paper presents the statistical results of an investigation on the willingness to accept COVID19 Booster vaccination. The results of the paper have been obtained by the responses to a questionnaire to inquire also. The study is well organized and it is based on real data. The results on the response can be useful to risk managers in charge of organizing vaccination programs.

Therefore, I recommend publication after some revisions aimed at referring their investigation also to a general framework of the complex dynamics of the virus in human societies. In details, the authors should be invited to care about the following topics:

Thank you for your constructive comments and careful review. We have prepared and presented the responses to the reviewer’s comments in a point-by-point manner below.

A paper on a closed topic has been recently published on Vaccines. The authors should consider this paper and discuss similarities and differences.

Folcarelli, L., Del Giudice, G.M., Corea, F., Angelillo, I.F., Intention to Receive the COVID-19 Vaccine Booster Dose in a University Community in Italy, Vaccines 10(2), 146.

Thank you for your valuable comment. We completely agree with your opinion. Your suggested article and our study are comparable; both emphasize that age is associated with vaccine hesitancy. Therefore, we have included it in the manuscript and cited it as a reference in the INTRODUCTION and DISCUSSION sections.

“This result was consistent with previous reports that younger individuals are less likely to accept vaccines in the primary vaccination series as well as the booster dose [5,13,24-27].” [Addressed in Page No 6 Line No 191 to 193]

The use of the statistics to different aspects related to the interpretation of the contents of the paper is mainly left to the authors’ claim, but the authors should make an effort to address the use which goes beyond crisis management, but also to various aspects related to the pandemics viewed as a complex systems in a globally connected world. In particular, there are studies generated by the following article:

https://www.worldscientific.com/doi/pdf/10.1142/S0218202520500323

which investigates the role of the immune competition on the shape of subsequent waves after the first wave.

This role is activated by vaccination programs etc.

Thank you for these valuable comments and informing us of these important studies. We have cited the study in the statement discussing the immune competition from a global point of view as follows:

“Finally, considering the development of COVID-19 vaccination strategies, vaccine equity and immune competition are also key determinants [40].” [Addressed in Page No 6 Line No 219 to 221]

Reviewer 2 Report

This is an interesting study although it has some major drawbacks as well as minor issues that I’m listing below.

Methodological issues:

  1. What type of vaccine was received by the studied participants? Please clarify.
  2. When were the serological tests performed? What time from the last dose of the vaccine? Was it homogenic for all participants? Please clarify.
  3. Line 105-106: „Children aged ≥12 years were allowed to receive the 105 COVID-19 vaccination; thus, we included 58 participants aged 12–19 years.” – Why did you include participants <18 years? There are only 58 subjects among nearly 2500 participants, not enough for enough statistical power.
  4. Line 108-115: this paragraph is not well-phrased, please revise for clarity.

Introduction:

  1. Line 49-50:” including the BNT162b2, mRNA-1273 Moderna vaccine, Janssen Ad26.COV2.S, and Sinopharm inactivated virus vaccines” – Chaos in names of vaccines, please provide a code name (e.g., Ad26.COV2.S) and manufacturer in the brackets (Janssen/Johnson&Johnson).
  2. Line 56-57: „The Centers for Disease Control and Prevention has recommended a third 56 COVID-19 vaccine dose in immunocompromised individuals [9]” – please be clear to distinguish additional vaccine dose for immunocompromised from the booster dose for all. 
  3. Line 63-54: „However, information on the factors affecting COVID-19 booster vaccine hesitancy among fully-vaccinated individuals is limited.” – there are more and more works on this, including those published in the Vaccines; please refer and include in the Discussion:
  • https://doi.org/10.3390/vaccines9111286
  • https://doi.org/10.3390/vaccines9121424
  • https://doi.org/10.3390/vaccines9121437
  • https://doi.org/10.3390/vaccines9111358

Results

  1. Line 145 – 147: „Among the 2439 participants with a mean (±SD) age of 52.6±18.9 years, median IgG-S antibody titer of 324.9 (AU/mL), and of whom 41.7% were male, 97.9% indicated their intent to accept the third vaccination dose.” – This is great, but it means that ONLY 51 PARTICIPANTS from nearly 2,5 thousand did not want to receive a booster. Statistically speaking this is challenging to analyze.
  2. Line 164 – 167: „Concerns about adverse reactions and efficacy of the vaccine were the top reasons to hesitate a third vaccination. In cases where the respondents  were not concerned about efficacy or adverse reactions, these factors were the top reasons for accepting the third vaccination.”  - Second sentence is unclear, please rephrase.
  3. „Participants included healthcare workers,  frontline workers, administrative officers, general residents, and residents of long-term care facilities” – did the group vary regarding acceptance and fears over a booster dose? If not, please state, but maybe they did vary? Clarify.
  4. Did the participants receive different types of COVID-19 vaccines? If so, please analyze whether this was a factor differentiating the acceptance of the booster.

Discussion

  1. Line 196-198: „Approximately 97.9% of FVCS participants who had their own antibody titers measured accepted a third vaccination. The estimated global acceptance rate of the COVID-19  vaccine was reported to be at 73.2% [22], which is lower than our results.” – The study you refer to is about the general acceptance, not acceptance of the booster dose. Please refer to the specific studies, see my remark for the Introduction for reference suggestions.

Keywords

  1. Please add „pandemic”

Author Response

Response to reviewer 2

This is an interesting study although it has some major drawbacks as well as minor issues that I’m listing below.

We greatly appreciate your careful review. We have prepared and presented the responses to the reviewer’s comments in a point-by-point manner as follows:

Methodological issues:

What type of vaccine was received by the studied participants? Please clarify.

When were the serological tests performed? What time from the last dose of the vaccine? Was it homogenic for all participants? Please clarify.

Thank you for these valuable comments. We completely agree with the reviewer; we have revised the text in the main manuscript as follows:

A total of three participants received two doses of mRNA-1273 and all the others received two doses of BNT162b2. All the participants who received mRNA-1273 were non-healthcare workers. The vaccination dose was the same for all the participants. The blood sampling was performed between September 8 and October 8, 2021. The mean (SD) of the interval days between second dose vaccination and serological tests were 180.5 (±32.2) days. Participants were a predominantly Japanese homogenic group, recruited mainly from rural Fukushima Prefecture, including Ishikawa gun, Soma City, and Minamisoma City.[Addressed in Page No 3 and 4 Line No 94 to 101]

Line 105-106: „Children aged ≥12 years were allowed to receive the 105 COVID-19 vaccination; thus, we included 58 participants aged 12–19 years.” – Why did you include participants <18 years? There are only 58 subjects among nearly 2500 participants, not enough for enough statistical power.

Thank you for these valuable comments. We analyzed the participants excluded under 18 years old, and found no difference between them and all the participants in the analysis. The results are shown below. On the other hand, information on the participants in the 12–18-years old groups were limited. Therefore, after careful discussion, we decided to include all the participants in the analysis.

B (se)

OR (95% CI)

p-value

Age

0.010

0.98 (0.96–1.00)

0.017

Sex (base: male)

0.321

1.07 (0.60–1.93)

0.81

Number of whole-body adverse reaction

0.101

1.04 (0.86–1.26)

0.68

IgG antibody titer

0.894

2.57 (1.30–5.08)

0.007

Municipality (base: Hirata)

Soma

0.210

0.43 (0.17–1.12)

0.099

Minamisoma

0.207

0.57 (0.28– 1.16)

0.122

Line 108-115: this paragraph is not well-phrased, please revise for clarity.

Thank you for these valuable comments. We completely agree with the reviewer; we have revised the corresponding text in the main manuscript as follows.

“We conducted a questionnaire survey on the acceptance of the third vaccination. The questionnaire items concerning the reasons for acceptance or hesitance towards the third vaccination could be answered in multiple ways. The papers on vaccine hesitancy [10,11] were used as reference for preparing the questionnaire survey. We also considered the opinion of the medical staff, local government staff, and researchers involved in the vaccination process.” [Addressed in Page No 3 Line No 111 to 116]

Introduction:

Line 49-50:” including the BNT162b2, mRNA-1273 Moderna vaccine, Janssen Ad26.COV2.S, and Sinopharm inactivated virus vaccines” – Chaos in names of vaccines, please provide a code name (e.g., Ad26.COV2.S) and manufacturer in the brackets (Janssen/Johnson&Johnson).

Thank you for these valuable comments. We completely agree with the reviewer; we have revised the text in the main manuscript as follows.

“Many efforts have been made to develop vaccines, including the BNT162b2 (Pfizer/BioNTech), mRNA-1273 (Moderna), Ad26.COV2.S (Janssen/Johnson&Johnson), and BBIBP-CorV (Sinopharm), which were approved by the World Health Organization”

[Addressed in Page No 2 Line No 48 to 51]

Line 56-57: „The Centers for Disease Control and Prevention has recommended a third 56 COVID-19 vaccine dose in immunocompromised individuals [9]” – please be clear to distinguish additional vaccine dose for immunocompromised from the booster dose for all.

Thank you for these valuable comments. We completely agree with the reviewer; we have revised the text in the main manuscript as follows.

“The Centers for Disease Control and Prevention has recommended a third COVID-19 vaccine dose in immunocompromised individuals [10]. The booster dose is the same for the BNT162b2 and half for the mRNA-1273 compared to the dose in the primary vaccination series.”

[Addressed in Page No 2 Line No 56 to 59]

Line 63-54: „However, information on the factors affecting COVID-19 booster vaccine hesitancy among fully-vaccinated individuals is limited.” – there are more and more works on this, including those published in the Vaccines; please refer and include in the Discussion:

https://doi.org/10.3390/vaccines9111286

https://doi.org/10.3390/vaccines9121424

https://doi.org/10.3390/vaccines9121437

https://doi.org/10.3390/vaccines9111358

Thank you for these valuable comments and providing these important references. We have cited your suggested paper in the introduction and discussion sections.

Results

Line 145 – 147: „Among the 2439 participants with a mean (±SD) age of 52.6±18.9 years, median IgG-S antibody titer of 324.9 (AU/mL), and of whom 41.7% were male, 97.9% indicated their intent to accept the third vaccination dose.” – This is great, but it means that ONLY 51 PARTICIPANTS from nearly 2,5 thousand did not want to receive a booster. Statistically speaking this is challenging to analyze.

Thank you for these valuable comments. We completely agree with your opinion that the number of people who didn’t want to receive a booster dose is small, which was challenging for the statistical analysis. Therefore, we added the following sentence to the limitation. However, it is also important to know the factors associated with vaccine hesitancy, among the population where many people accepted the third dose. Thus, we analyzed the data by limiting the number of variables in the logistic model.

Besides, a few people hesitated towards the third vaccination, which made the statistical analysis challenging.[Addressed in Page No 6 Line No 213 to 215]

Line 164 – 167: „Concerns about adverse reactions and efficacy of the vaccine were the top reasons to hesitate a third vaccination. In cases where the respondents were not concerned about efficacy or adverse reactions, these factors were the top reasons for accepting the third vaccination.” - Second sentence is unclear, please rephrase.

Thank you for these valuable comments. We completely agree with the reviewer’s comments; we have revised the text in the main manuscript as follows.

“Concerns about adverse reactions and efficacy of the vaccine were the top reasons for hesitance towards a third vaccination. In contrast, “Necessary for infection control” was the top reason for acceptance towards the third vaccination in addition to lack of concern about the efficacy and adverse reactions.” [Addressed in Page No 4 Line No 165 to 168]

Participants included healthcare workers, frontline workers, administrative officers, general residents, and residents of long-term care facilities” – did the group vary regarding acceptance and fears over a booster dose? If not, please state, but maybe they did vary? Clarify. Did the participants receive different types of COVID-19 vaccines? If so, please analyze whether this was a factor differentiating the acceptance of the booster.

Thank you for your valuable insights. We completely agree with your opinion that it is important to examine whether the occupation type was associated with vaccine acceptance. However, we were unable to collect data on the occupation. Thus, we have added the following statement to the limitations.

“Moreover, we were unable to collect information on the occupations that might affect vaccine acceptance. Furthermore, nearly everyone had received BNT162b2, making it difficult to verify whether vaccine acceptance differed between the different types of vaccines.”

[Addressed in Page No 6 Line No 216 to 219]

Discussion

Line 196-198: “Approximately 97.9% of FVCS participants who had their own antibody titers measured accepted a third vaccination. The estimated global acceptance rate of the COVID-19 vaccine was reported to be at 73.2% [22], which is lower than our results.” – The study you refer to is about the general acceptance, not acceptance of the booster dose. Please refer to the specific studies, see my remark for the Introduction for reference suggestions.

Thank you for these valuable comments. We completely agree with the reviewer, and have revised the text in the main manuscript as follows.

“Approximately 97.9% of FVCS participants who had their own antibody titers measured accepted the third vaccination. This result was higher than that of other reports on third vaccination acceptance [13-15]” [Addressed in Page No 6 Line No 198 to 200]

Keywords

Please add „pandemic”

Thank you for these valuable comments. We have added it to the Keywords.

Round 2

Reviewer 2 Report

I thank Authors for providing the responses to my comments and foremost, for the revision. There are however some further points which need to ba addressed

  1.  Thank you for adding an information on CDC recommendations. However, you need to distinquish ADDITIONAL dose for immunocompromised patients from BOOSTER dose for everyone. In other words, initial vaccination protocol includes two doses of mRNA vaccine, while for immunocompromised subjects it includes three doses, with a third dose given 21-28 days after a second. This additional dose is not equivalent to a booster shot, given 5-6 months after completing an initial vaccine protocol. In immune deficient subjects, a booster shot (in case of mRNA vaccine) is actually a FOURTH dose. Please be clear on this matter in your manuscript.  Consult a WHO nomenclature: https://www.who.int/news/item/22-12-2021-interim-statement-on-booster-doses-for-covid-19-vaccination---update-22-december-2021
  2. The low sample size of individuals aged <18 should be acknowledged in the Discussion within the study limitations section.
  3. When reporting mean/SD time from a vaccine dose to serological test, please add a range. Could the difference in this time affect your results? Consider discussing it.
  4.  When providing the manufacturer names for particular vaccine type, please add city (state in case of US) and country(countries) - otherwise, MDPI editors will ask you to do so at proof-reading stage.
  5. Maybe change a word "top" to "main" or "most frequent" when discussing the most frequent reasons of third dose hesitancy.

Author Response

Response to reviewer 2

I thank Authors for providing the responses to my comments and foremost, for the revision. There are however some further points which need to ba addressed.

Thank you for your constructive comments and careful review. We have prepared and presented the responses to the reviewer’s comments in a point-by-point manner below.

Thank you for adding an information on CDC recommendations. However, you need to distinquish ADDITIONAL dose for immunocompromised patients from BOOSTER dose for everyone. In other words, initial vaccination protocol includes two doses of mRNA vaccine, while for immunocompromised subjects it includes three doses, with a third dose given 21-28 days after a second. This additional dose is not equivalent to a booster shot, given 5-6 months after completing an initial vaccine protocol. In immune deficient subjects, a booster shot (in case of mRNA vaccine) is actually a FOURTH dose. Please be clear on this matter in your manuscript. Consult a WHO nomenclature: https://www.who.int/news/item/22-12-2021-interim-statement-on-booster-doses-for-covid-19-vaccination---update-22-december-2021

Thank you for these valuable comments. We have revised the introduction section as follows.

“It was reported that six months after the administration of the second dose of the BNT162b2 vaccine, the humoral response was substantially reduced [8]. In several jurisdictions, booster vaccination has been authorized by regulatory authorities [9]. The booster dose is the same for the BNT162b2 and half for the mRNA-1273 compared to the dose in the primary vaccination series.”

The low sample size of individuals aged <18 should be acknowledged in the Discussion within the study limitations section.

Thank you for these valuable comments. We have added following sentence in the limitation section.

Besides, a few people hesitated towards the third vaccination and only 58 of the approximately 2,500 participants were under the age of 18, which made the statistical analysis challenging.

When reporting mean/SD time from a vaccine dose to serological test, please add a range.

Thank you for pointing that out. We corrected it as noted.

Could the difference in this time affect your results? Consider discussing it.

Thank for you comment. I agree with you that it is important to consider the relationship between the time from a vaccine dose to serological test and vaccine hesitancy. However, the purpose of this study was to examine the effect of the antibody values returned to the individual on their vaccination intentions. Antibody titers were affected by days. (Correlation coefficient was -0.3832) Therefore, we believe that including both variables in the logistic model is problematic from the perspective of multicollinearity. the purpose of this study was to examine the effect of the antibody values returned to the individual on their vaccination intentions. For this reason, the final model includes only one of the two, i.e., only the value of the antibody. We included the following text in the METHOD.

“Since the antibody titers were affected by days from the vaccination (Correlation coefficient was -0.38), we used only the antibody titer as the final model from the perspective of multicollinearity.”

When providing the manufacturer names for particular vaccine type, please add city (state in case of US) and country(countries) - otherwise, MDPI editors will ask you to do so at proof-reading stage.

Thank you for pointing that out. We corrected it as noted.

Maybe change a word "top" to "main" or "most frequent" when discussing the most frequent reasons of third dose hesitancy.

Thank you for pointing that out. We corrected it as noted.